# Inside the decentralised casino: A longitudinal study of actual cryptocurrency gambling transactions

**Oliver J. Scholten**⊙*, **David Zendle, James A. Walker**

Department of Computer Science, University of York, York, Yorkshire, United Kingdom

\* ojs524@york.ac.uk

## Abstract

Decentralised gambling applications are a new way for people to gamble online. Decentralised gambling applications are distinguished from traditional online casinos in that players use cryptocurrency as a stake. Also, rather than being stored on a single centralised server, decentralised gambling applications are stored on a cryptocurrency's blockchain. Previous work in the player behaviour tracking literature has examined the spending profiles of gamblers on traditional online casinos. However, similar work has not taken place in the decentralised gambling domain. The profile of gamblers on decentralised gambling applications are therefore unknown. This paper explores 2,232,741 transactions from 24,234 unique addresses to three such applications operating atop the Ethereum cryptocurrency network over 583 days. We present spending profiles across these applications, providing the first detailed summary of spending behaviours in this technologically advanced domain. We find that the typical player spends approximately $110 equivalent across a median of 6 bets in a single day, although heavily involved bettors spend approximately $100,000 equivalent over a median of 644 bets across 35 days. Our findings suggest that the average decentralised gambling application player spends less than in other online casinos overall, but that the most heavily involved players in this new domain spend substantially more. This study also demonstrates the use of these applications as a research platform, specifically for large scale longitudinal in-vivo data analysis.

**Data Availability Statement:** All files are available in the OSF repository available at https://osf.io/8bfyj/.

**Funding:** This work was supported by the EPSRC Centre for Doctoral Training in Intelligent Games & Games Intelligence (IGGI) [EP/L015846/1] (OJS,

## Introduction

Decentralised gambling applications are a new form of online gambling which use cryptocurrency technology to process payments and calculate game outcomes [1]. These applications vary in terms of the games they provide, and the cryptocurrencies they use. This work focuses on simple casino type games of chance, like dice rolls and coin flips, available through several applications operating atop the Ethereum cryptocurrency network. The Ethereum network is the oldest and most popular by market capitalisation of cryptocurrency networks which explicitly support smart contracts (see https://www.coinbase.com/, accessed 12/11/2019). These contracts, which are computer programs, are the core technology enabling these applications [2].

PhD Scholarship) and the Digital Creativity Labs (digitalcreativity.ac.uk) (JAW, Research Fellow), jointly funded by EPSRC/AHRC/Innovate UK under grant no. EP/M023265/1. The funders had no role in study design, data collection and analysis, decision to publish, or preparation of the manuscript.

**Competing interests:** The authors have declared that no competing interests exist.

Understanding the behaviours of users of these applications is important for understanding how this technology is affecting the public in comparison with existing online gambling platforms [3]. It is also important for understanding the prevalence of patterns of problematic spending [4] among its users. No existing work has contributed to understanding decentralised gambling application players using their transaction data. This work begins addressing this gap in the literature by analysing three such applications operating atop the Ethereum network. We begin by briefly describing relevant existing work in this domain.

## Background

Player behaviour tracking is a subset of gambling research which aims to better understand how people gamble using actual betting data. Historically, this field has been limited by the availability of large scale, real-life observational data [5] given its commercial sensitivity and personal nature. On top of this, many existing studies have only had access to daily aggregate data, as opposed to individual transaction level data. This means that although methods do exist specific to more granular data, for example Fiedler's work on poker play [6], little exists specific to casino game play. The use of cryptocurrencies for gambling challenges this *status quo*, offering data access at previously inconceivable granularity. This access invites analysis of player spending in this new domain, as this paper explores.

All transactions of cryptocurrencies such as Bitcoin and Ethereum are recorded on public ledgers known as blockchains. Decentralised gambling applications involve the wagering of cryptocurrencies. When individuals place wagers using a decentralised gambling application, their transactions are therefore recorded on these public ledgers. This means every transaction to and from these applications is publicly available. This represents a paradigm shift in terms of data availability for gambling researchers, and invites a new branch of player behaviour tracking research focused on the use of this data for understanding player spending. Furthermore, the psuedo-anonymous nature of cryptocurrency transactions means the data is publicly available in an already anonymised form, mitigating many of the limitations associated with the use of personally identifiable information for academic research. These factors, public availability and pre-anonymisation, mean decentralised gambling applications may find use at the heart of data-driven gambling research in the future, over industry collaborations commonly found in existing literature [5, 7].

Existing work on decentralised (gambling) applications of any kind has been limited [7, 8]. Early work by Gainsbury [1] describes their existence and potential to revolutionise the provision of gambling services online. However, little has been done to capitalise on their data transparency and public availability described above. Literature on the analysis of cryptocurrency transactions in general terms has also been sparse, with none to the authors' knowledge exploring their use for gambling research specifically. This paper is the first to explore such applications through the lens of gambling studies, using established methods to examine player spending.

Literature exploring the use of online gambling transaction data for gambling research however, does exist [5], and provides a foundation upon which methods of exploration can be built. A collection of behavioural measures have been described by the first series of papers to explore online gambling. This began with LaBrie *et al.*'s 2007 study on the gambling behaviours of sports bettors [9]. In this study, temporally oriented measures such as duration and frequency of play, financially oriented measures such as mean bet size and total expenditure, and loss oriented measures like net loss and percentage loss (of total amount wagered) were calculated. They found, using descriptive statistics across cohorts of players, that differences in behavioural measures between cohorts existed, and that an empirically determined group of

heavy bettors spent more, and more frequently, across both fixed odds and live action betting. In a similar vein, a further paper by LaPlante *et al.* explored the individual behaviours of poker players, again calculating behavioural measures using their transaction data [10]. Presenting descriptive statistics in a similar way, they were able to provide a baseline further researchers such as Fiedler [11] could build upon, extending our knowledge of online gambling in general, and in this case poker play in particular. Finally, and most relevantly, LaBrie *et al.* computed identical measures to their earlier paper [9], instead applying them to casino game players. Once again, differences were found between cohorts of players using an empirically determined split (95%:5% by total amount wagered), laying a foundation for developing an understanding of gambling behaviour in a previously unknown domain.

As noted earlier in this section, prior work has established baseline measurements of various behavioural factors in domains as diverse as poker play and sports betting. However, no such quantification of player behaviour in the crypto-gambling domain exists. Replicating this analysis on new cryptocurrency data would help establish a baseline from which further research could be conducted, and offer novel insights into player behaviours in this technologically sophisticated domain. This also addresses several recent reviews which have called for work to develop our understanding of the use of new technologies for gambling [5, 12].

Finally, work applying more advanced analytical tools to player transaction data, such as Percy *et al.*'s use of supervised machine learning models [13] to predict self-exclusion, and Philander's exploration of data mining procedures [14] to identify high-risk gamblers, each build on the measures calculated in the papers described in this section. Establishing a descriptive baseline is therefore an important first step in the development of more advanced analytical algorithms.

## Hypotheses

Previous work on the analysis of *in vivo* gambling transactions has varied between game type and cohort characteristics [5]. As none have focused on the use of decentralised applications for gambling of any kind—except work by Gainsbury [1], which used high level usage statistics —we expect to find gambling behaviours consistent with analyses on casino game players whose gameplay is most similar to that of the applications described in detail below. We include a table from one such study by Labrie et al [9] in the appendix for quick comparison. Secondly, we do not expect the data gathered directly from the Ethereum cryptocurrency network to be usable for player behaviour research without first applying some data cleaning methods. Of highest importance is the potential for the presence of non-human players, known as bots, in the data set. Bots may exist here for a number of reasons—for example, to artificially inflate the perceived popularity of the applications they are transacting with, or to attempt to win the jackpot from an application once it becomes statistically worthwhile to pursue. We cannot infer the reasoning behind bots' existence, but can build evidence to identify their presence by assessing how much typical 'player' behaviour from each game deviates from those of other similar games.

## Present study

This work describes the behaviour of a large cohort of decentralised gambling application users over a 583 day period, spanning from the creation of each application's smart contracts up until the 9th March 2020 (see Figs 1 and 2). By using cryptocurrency transaction data gathered directly from the Ethereum cryptocurrency blockchain we are able to calculate behavioural measures using individual bet level data as opposed to aggregates of any kind, e.g. daily/weekly. Behavioural measures, including descriptions of the typical (median) player of

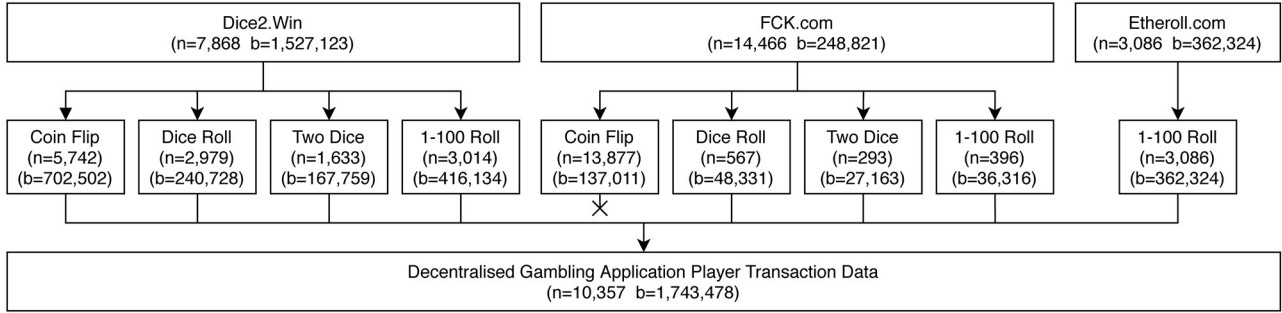

**Fig 1. Provider-game combinations, including unique address and bet counts taken forward to the final player transaction set.**

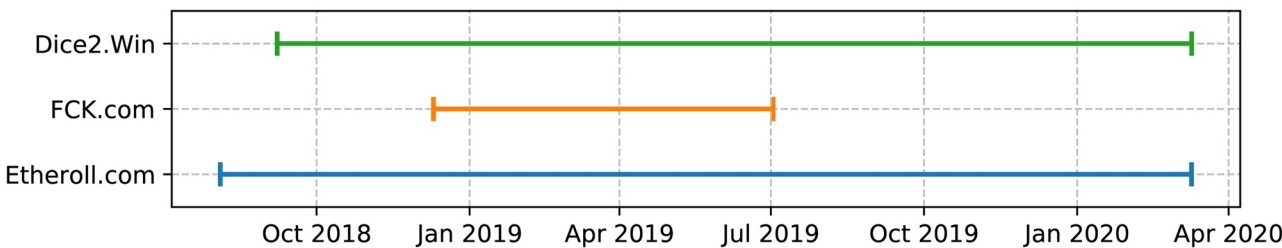

**Fig 2. Transaction data gathering timelines for each of the three decentralised gambling applications studied.**

each of the games available through each of the applications is described. We perform four distinct analyses following identification of likely non-human players: (i) a statistical comparison between human and bot players' behavioural measures; (ii) an epidemiological description of the gambling behaviour of (human) players of decentralised gambling applications; (iii) a statistical assessment of the relationships between existing behavioural measures of players in this new domain, and (iv) an epidemiological description of the gambling behaviours of empirically-determined heavily involved players as found in LaBrie *et al.*'s original work [4].

## Materials and methods

Written ethical approval for this study was granted by the Physical Sciences Ethics Committee at the University of York, application reference: Scholten021219.

### Data sample

Data gathered for this study includes transactions to and from three decentralised gambling applications operating atop the Ethereum cryptocurrency network. These applications were selected based both on their rank on an officially recognised application ranking service StateOfTheDApps, available at https://stateofthedapps.com, and on the subjective technical simplicity of their smart contracts. This simplicity is dependent on the author's understanding of the Solidity programming language, as encoded transactions to these contracts require decoding in order to extract the sizes of bets and player outcome selections. A deeper understanding of the language these contracts are written in would increase the number of applications that could be analysed. However, given the youth of this technology, our goal here is to first understand a small sample.

**Table 1. Smart contract addresses for each decentralised gambling application used in this study.**

| Provider | Address |
|---|---|
| Dice2.Win | 0xD1CEeeeee83F8bCF3BEDad437202b6154E9F5405 |
| Etheroll.com | 0xA52e014B3f5Cc48287c2D483A3E026C32cc76E6d |
| FCK.com | 0x999999C60566e0a78DF17F71886333E1dACE0BAE |

**Table 2. Meta data for each application gathered as part of this study.** Bet and Payout values are given in ETH, and starting and ending blocks and dates represent the time window from which transactions were gathered. All transaction data used in this study is available at https://osf.io/8bfyj/.

| | Etheroll.com | FCK.com | Dice2.Win |
|---|---|---|---|
| Unique Users | 3,086 | 14,466 | 7.868 |
| Games | 1 | 4 | 4 |
| Bet Value | 420,942.442 | 465,195.853 | 1,267,239.951 |
| Payout Value | 419,067.602 | 462,136.712 | 1,245,815.279 |
| Start Block | 6084746 | 6859200 | 6287216 |
| End Block | 9638617 | 8071084 | 9639151 |
| Start Date | 2018-08-04 04:27:21 | 2018-12-10 06:05:13 | 2018-09-07 08:17:20 |
| End Date | 2020-03-09 17:35:39 | 2019-07-02 08:49:06 | 2020-03-09 19:32:55 |

The first such application is Etheroll, described as '*an Ethereum smart contract for placing bets on [a] provably-fair dice game using Ether with no deposits or sign-ups. Each dice roll is provably random and cryptographically secure thanks to the nature of the Ethereum blockchain*' (description taken from https://etheroll.com/#/about). The second application is Dice2Win, which offers both single and double dice rolls, coin flips, and 1-100 rolls, all through a smart contract in the same way as the Etheroll application. Finally, the FCK application offered a collection of simple casino games such as roulette, guess-the-suit, guess-the-number, etc. The FCK application ceased operation on the 8th July 2019, yet with 349,195 transactions since its creation (December 10th 2018) it provides substantial data for the present study, and implements a (technically) simple contract in terms of transaction decoding. The Ethereum smart contract addresses associated with each of these applications at the time of data gathering are presented in Table 1. Summary statistics of the data collected from these applications is presented in Table 2.

## Data cleaning

Transactions to and from the contracts associated with each of the applications, gathered from the start of their operations until 9th March 2020 (see Fig 2), yield a total of 2,232,741 bets originating from 24,234 unique addresses. Of these addresses, 14,466 transacted with the FCK application, a further 7,868 with the Dice2Win application, and a final 3,086 with the Etheroll application (see Fig 1). Fig 3 plots the cumulative value of the bets placed both in each application alone, and combined across the duration of this study.

The transaction data for each of these applications was gathered using the Etherscan API, which offers an interface through which transactions on the Ethereum blockchain can be directly inspected. The Etherscan API can be found at https://etherscan.io.

As the raw dataset is publicly available via the Ethereum blockchain, the data repository associated with this work contains the matched bets used to calculate the measures below in an accessible format (CSV). This data includes the hashes (unique identifiers) of both the bet placement and payout transactions such that the sums of the costs to and from each unique

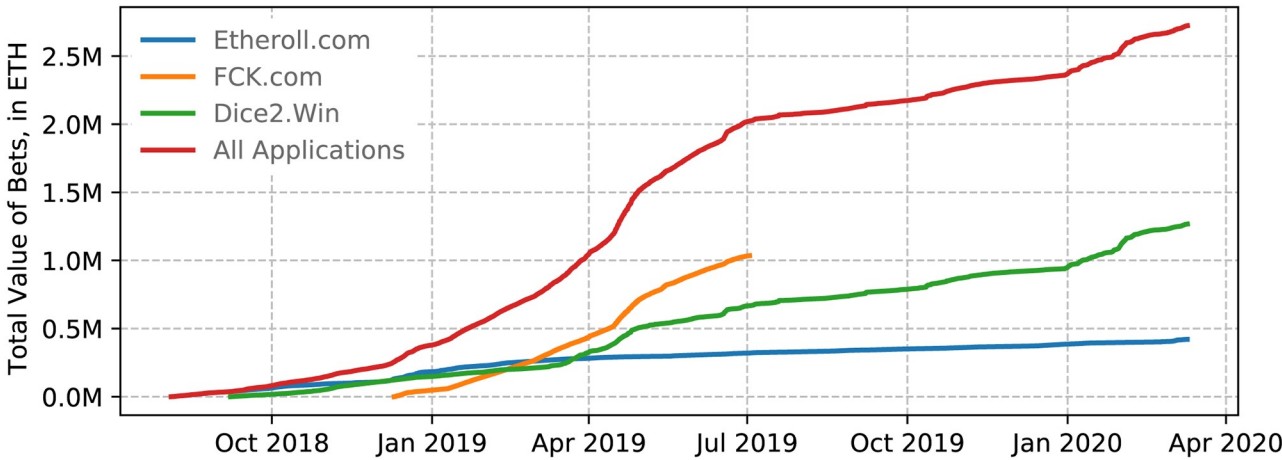

**Fig 3. Cumulative value of bets placed through each application individually, and all applications combined over the period studied.** The data and code used to create this figure is available at https://osf.io/8bfyj/.

address can be verified. The transaction data used for this study are available in full at https://osf.io/8bfyj/.

## Measures

The psuedo-anonymous nature of the cryptocurrency transactions from which the present data set was drawn mean that no demographic information such as age, gender, or income, is available for any of the unique cryptocurrency addresses in the set. As such, no demographic analysis was possible—this aligns with existing literature where demographic data was not found to be of particular interest in comparison to behavioural measures based on the transaction data alone [9].

The variables computed as part of this study are based on those calculated by LaBrie *et al.*'s seminal investigation into internet casino games. These include the duration of betting, which is calculated as the time elapsed (in days) between the placement of the first bet and the placement of the last. This is rounded up to the nearest day in cases where bets were made across a midnight boundary, for example, the placement of bets both at 22:00 on a given day and again at 09:00 on the following day, are counted as having a duration of two days even though they are within 24 hours of one another. Using this we could compute the frequency of betting activity by taking the total number of days in which one or more bets was placed and dividing it by the duration of betting. This yields a percentage, with value of 100% equating to betting every day for the known duration of the use of the decentralised gambling application.

As in the original work, we calculated the average bets per day by dividing the total number of bets made by each player, by the total number of days on which a bet was placed (as used when computing the frequency above). The total amount wagered (in ETH) for each player is also retrieved, along with the total losses they incurred (also in ETH), from which their net loss is calculated. Finally, the percentage loss for each player is determined by dividing the net loss by the total amount wagered, and multiplying by 100. As in LaBrie *et al.*'s original work, the large sample size (n = 23,365) of the players of the three decentralised gambling applications gathered in this work mean that the practical significance of any statistical differences between any of the measures calculated may be limited.

In order to promote reproducibility in our work, and to encourage further studies in this domain, the code used to calculate these measures across each of the unique addresses is

available as part of the gamba library ([www.gamba.dev](www.gamba.dev)). This library also contains methods capable of exactly replicating LaBrie *et al.*'s original work, plus each of the computations required to replicate all tables in the present study. The publication of the complete data set and fully documented analytical code is a core contribution of this paper.

## Results

### Non-human players

Before presenting descriptive statistics for cryptocurrency gamblers, we must first ensure that the transactions used originate from human players. Given the lack of established methods in making this distinction, a naïve approach, inspired by LaPlante *et al.*'s use of the Kolmogorov-Smirnov test [10], is to quantify the differences between the distributions of each of the behavioural measures for players across each of the games. We reason that if the majority of unique addresses' transactions originate from human players, collections of addresses transactions' which deviate significantly from this norm may be non-human in origin. This reasoning finds support in the fictitious scenario where an auto-betting algorithm with few parameters is used by many accounts, as this would create groups of behaviourally similar transaction sequences which would stand out. Fig 4 illustrates this theory, with a smaller peak indicating human players in a population with non-human players, and a second peak indicating non-human player behaviours.

To this end, we first split the collection of all gathered transactions by application, and then again by game. This resulted in 9 distinct transaction sets, each for a single application-game combination—for example; coin-flip players on the dice2.win application, two-dice players on the fck.com application, etc. The dice2.win and fck.com applications each offer 4 games, plus etheroll.com's 1-100 roll, yields 9 different games in total. From here, a two sample Kolmogorov-Smirnov test (K-S II)—which quantifies the likelihood that two samples have been drawn from the same distribution—was computed for each pair of measures, across each of the applications. This resulted in a 9x9x8 matrix of coefficients, with axes; application-game combinations (9), application-game combinations again (9), and behavioural measures (8). Algorithm 1 shows the design of this pairwise behavioural measure comparison, a Python implementation of which is available at [www.gamba.dev](www.gamba.dev). It should be noted that performing this many K-S II tests without correction limits their individual descriptive power. That considered, the uncorrected coefficients of these tests can still be used to broadly assess differences between the distributions.

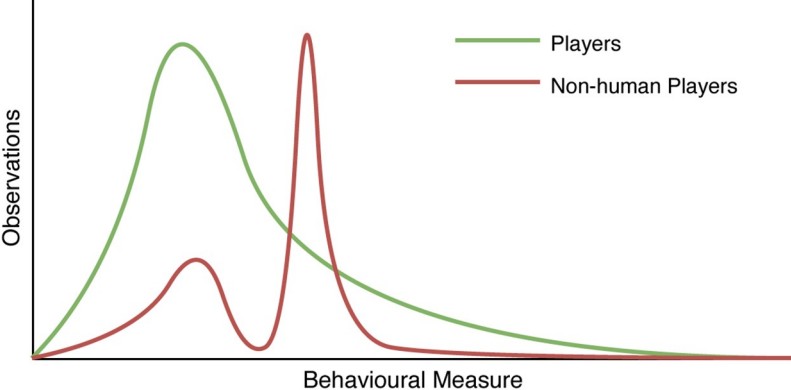

**Fig 4. Theoretical differences in distributions of behavioural measures between human and non-human players.** The second spike is created when multiple addresses transact in the same way, e.g. using by using computer code.

[https://doi.org/10.1371/journal.pone.0240693.g004](https://doi.org/10.1371/journal.pone.0240693.g004)

**Algorithm 1:** Two sample Kolmogorov-Smirnov tests for each behavioural measure between groups of players, where each group represents the players of a single game on a single application. The Python implementation used in this study is available as part of the gamba library at www.gamba.dev.

```
Data: Behavioural measures for all players
Result: K-S II tests between measures for each application-game
combination
data = [player measures for each app-game combination];
allMeasureTests = [];
for measure in measures do
  // 2D matrix for one measure;
   testResults = [];
   for column in data do
     for row in data do
        testResults.append(KStest(column, row));
     end
   end
   allMeasureTests.append(testResults);
 end
```

Table 3 shows a single slice of this coefficient matrix corresponding to the behavioural measure of *duration* for each application-game combination described in Fig 1. From this slice alone it is clear that the coin-flip game on the fck.com application stands out against almost all others in terms of the size of the K-S II coefficient. Following the scipy library's description of the K-S II test; *if the K-S statistic is small or the p-value is high, then we cannot reject the hypothesis that the distributions of the two samples are the same*, it is likely that players of the fck.com coin-flip game are not similar to those of almost any other game. It is therefore possible that if the players of the other games are human, then fck.com coin-flip players are not. The results of these tests across each of the behavioural measures in the matrix appears to indicate non-zero differences between the fck.com coin-flip players against players of all other provider-game combinations. Add to this that the fck.com coin-flip game amassed 13,877 unique players over it's lifespan of 209 days compared to 567, 293, and 396 players among its other three games, it appears unlikely that the majority of transactions to this game are human in origin.

**Table 3. Two sample Kolmogorov-Smirnov test results for player durations across all provider-game combinations.**

| Provider | | d2w | | | | fck | | | | eroll |
|---|---|---|---|---|---|---|---|---|---|---|
| | Game | cf | sd | dd | oh | cf | sd | dd | oh | oh |
| d2w | cf | - | | | | | | | | |
| | sd | 0.16 | - | | | | | | | |
| | dd | 0.24 [†] | 0.10 | - | | | | | | |
| | oh | 0.17 | 0.03 | 0.08 | - | | | | | |
| fck | cf | **0.55** [†] | **0.39** [†] | **0.46** [†] | **0.39** [†] | - | | | | |
| | sd | 0.22 [†] | 0.08 [†] | 0.05 | 0.05 | **0.43** [†] | - | | | |
| | dd | **0.40** [†] | 0.25 [†] | 0.16 [†] | 0.23 [†] | **0.59** [†] | 0.20 [†] | - | | |
| | oh | 0.34 [†] | 0.19 [†] | 0.10 [†] | 0.17 [†] | **0.54** [†] | 0.13 [†] | 0.07 | - | |
| eroll | oh | 0.09 | 0.12 | 0.19 [†] | 0.11 | **0.49** [†] | 0.14 [†] | 0.34 [†] | 0.27 [†] | - |

[†] denotes a significant result ($p < 0.01$) and coefficients greater than 0.35 are highlighted. Key: d2w = Dice2.Win, fck = FCK.com, eroll = Etheroll.com, cf = coin flip, sd = single dice roll, dd = double dice roll, oh = 1-100 roll.

The two sample Kolmogorov-Smirnov test results between the dice2win coin flip players and the fck.com double dice players are also higher than any other non fck.com pair. Yet with no other pairs indicating distributional differences with this group this may be an artefact of the choice of game, or may be coincidental given the number of tests conducted. In each case, this anomaly invites further exploration but is considered out of scope of the present study.

Under the assumption that each of the remaining provider-game pairs' transactions originate from human players—which we found no evidence to refute—we discarded the fck.com coin-flip transactions. This left 8 application-game combinations of interest, whose 10,357 unique players' behavioural measures—using 1,743,478 transactions—were combined into a single data set, as performed in existing work in gambling behaviour analysis. A graphical breakdown of these application-game combinations is provided in Fig 1. As with the matched transactions described above, the table of behavioural measures calculated for each unique address in this study is available through https://osf.io/8bfyj/.

### Cryptocurrency gambling behaviours

Table 4 presents the behavioural measures described in the Measures section above for the cohort of players in the remaining transaction set. The majority of the measures have heavily skewed distributions, which limits the descriptive power of the parametric statistics presented. This table therefore extends LaBrie *et al.*'s original metrics [3] by including the inter-quartile ranges of each of the measures, plus the coefficients of a one sample Kolmogorov-Smirnov test for normality as reported by LaPlante *et al.* [10].

We find that with a median duration of 1 day and frequency of 100%, the typical player of decentralised gambling applications bets in a non-commital, and non-intense way. This contrasts LaBrie et al's original findings on regular casino players, who with a median duration of 246 days and frequency of 7% bet across a much longer term. This contrast may be explained in part by the youth of the applications studied here. Add to this a median bet count of 11 and we may assume that this typical player would play for one short session on a single application and then cease play or move to another application. This considered, the inter-quartile range for the duration indicates a portion of players remaining engaged for over a week of play. Combine this with the inter-quartile ranges for both the frequency and number of bets measures and we observe a wide range of possible behaviours between the 25th and 75th percentiles of the sample, across the measures calculated. This sentiment is shared in the number of bets placed per betting day, which, with a median of 6 and IQR of 21, encapsulates a wide range of possible behaviours for the majority of the sample.

**Table 4. Gambling behaviour of 10,357 decentralised gambling application players including a one sample Kolmogorov-Smirnov (K-S) test for normality.**

| Metric | Mean | STD | Median | IQR | K-S |
|---|---|---|---|---|---|
| Duration (days) | 30 | 81 | 1 | 10 | 0.841 |
| Frequency (%) | 76 | 36 | 100 | 50 | 0.966 |
| Number of Bets | 168 | 992 | 11 | 62 | 0.841 |
| Mean Bets/Day | 23 | 48 | 6 | 21 | 0.841 |
| Mean Bet Size | 1.15 | 11.8 | 0.11 | 035 | 0.504 |
| Total Wagered | 213.77 | 2451.85 | 1.40 | 16.59 | 0.504 |
| Net Loss | 2.91 | 49.86 | 0.04 | 0.71 | 0.213 |
| Percent Loss | 10.9 | 112.1 | 5.3 | 52 | 0.548 |

All K-S test statistic values are significant at the $p < 0.01$ level, STD = standard deviation, IQR = inter-quartile range.

The top four behavioural measures also present the highest one sample K-S test statistics of all of the measures. This is most likely an artefact of the heavily skewed nature of these measures, with thorough investigations of outliers across each measure representing an interesting area of future work.

The financially oriented measures, including the ETH per bet and the total amount wagered show similar oddities to the results regarding duration and frequency. With a median bet size in ETH of 0.11 (approximately equivalent to $14, see https://www.coinbase.com/price/ethereum for exchange rate data used throughout this work) and total amount wagered of 1.40 ETH (approximately $200), the typical player's spending is high considering the short duration of play. The granular and longitudinal nature of the transaction data prepared as part of this work mean that questions surrounding this behaviour can be explored in greater detail in further work, but are not expanded upon here.

The most comparable measure presented here with other gambling activities is the net and percentage loss measures, which with median values of 0.04 (ETH) and 5.3% respectively indicate modest losses for the typical player. As with other financially oriented measures, when framed in terms of the median duration this equates to a loss of 5.3% of the total amount bet per day.

Unlike the top four measures presented, the financially oriented measures do not present such high K-S test statistics, so are likely drawn from less extreme distributions. That considered, with test statistics of 0.504, the ETH per bet and total amount wagered measures still cannot be effectively described using parametric methods. As such, the means and standard deviations for each of the measures are reported in line with existing literature, but in this domain do little to develop our understanding of typical transactional behaviour.

## Relationships between behaviours

As with previous work exploring the behavioural measures used in this work [9, 10], heavily non-normal distributions mean that rank-order correlations are preferred over their parametric equivalents. Table 5 presents Spearman rank-order correlation coefficients between all of the behavioural measures calculated for players of all games combined, excluding the fck.com application's coin-flip players.

Of particular interest in these coefficients are those of substantial magnitude, as highlighted. We find that, as expected, frequency is negatively correlated with duration—this makes sense as given a larger number of possible days on which to place a bet, the probability of a player not placing one on a given day naturally increases. The measure of duration does not appear

**Table 5. Non-parametric Spearman rank-order correlations between all behavioural measures for decentralised gambling application players.**

| Measure | Duration | Frequency | # Bets | Bets /day | Eth /bet | Total wagered | Net loss | % loss |
|---|---|---|---|---|---|---|---|---|
| Duration | - | | | | | | | |
| Frequency | **-0.89** | - | | | | | | |
| # Bets | 0.63 | -0.45 | - | | | | | |
| Bets/day | 0.35 | -0.19 | **0.93** | - | | | | |
| Eth/bet | 0.16 | -0.10 | 0.26 | 0.24 | - | | | |
| Total wagered | 0.53 | -0.39 | **0.84** | **0.78** | **0.72** | - | | |
| Net loss | 0.12 | -0.10 | 0.15 | 0.14 | 0.15 | 0.20 | - | |
| % loss | -0.10 | 0.06 | -0.15 | -0.12 | -0.07 | -0.14 | 0.67 | - |

All values are significant at the $p < 0.01$ level. Coefficients of magnitude greater than 0.70 are highlighted.

substantially correlated with any remaining measures, with moderate values for both number of bets and total amount wagered. These each loosely support notions that the longer an individual uses a decentralised gambling application, the more bets they will place and the greater their total amount wagered will become. These each also make logical sense in the context of the gambling games these applications present.

Apart from its correlation with duration, the measure of frequency does not appear to relate to any other measures in any substantial way. With a coefficient of 0.46, its correlation with the number of bets an individual makes also makes intuitive sense. The more frequently a player places bets, the more bets they are likely to place over their gameplay career.

The number of bets appears strongly correlated to both the number of bets per day and the total amount wagered for users of decentralised gambling applications. With a coefficient of 0.93—the strongest of all pairs—it is clear that the number of bets an individual places over their duration of play directly relates to the number of bets they are likely to place on a given day. The number of bets measure also relates strongly (0.84) to the total amount wagered. Unsurprisingly, the number of bets an individual places on a given day is also strongly correlated (0.78) with their total amount wagered. As with other relationships between measures, this makes intuitive sense in the context of gambling games but nonetheless contributes to establishing a baseline for human players of such games.

The final coefficient of interest, and that of most potential scientific significance, is that between the ETH per bet and the total amount wagered. With a reported coefficient of 0.78, our results suggest that those who place larger bets are more likely to wager larger total amounts over the duration of their betting careers. The implications of this finding are deferred to the discussion. However, this appears to suggest that this measure may be an important predictive indicator in the cryptocurrency domain. It may assist in terms of identifying the potential for financial harm via unsustainable spending among players—a finding in line with existing work in player behaviour tracking research [5].

Both the measures of net loss and percent loss do not appear meaningful in relation to the other measures reported in this work, so will not be discussed in detail. We now move on to report descriptive statistics regarding the most heavily involved bettors in the data set, and contrast them to the majority of low and moderately involved bettors.

## Heavily involved bettors

Heavy involvement by any of the behavioural measures used here may be detrimental to the individuals affected. For example, those most heavily involved in terms of the duration of their play will naturally have less time for other commitments, or those with large net losses may face financial repercussions should their income not support such expenditure. We explore heavy involvement with respect to total wagered, as it has the most obvious financial repercussions for the individuals in the cohort. This follows LaBrie *et al.*'s rationale for exploring the same measure in a cohort of casino gamblers. We include LaBrie et al's original figures for quick comparison in S1 Table, although such comparisons are heavily nuanced given the differences between decentralised and regular online casinos.

Table 6 presents each of the descriptive statistics for each of the behavioural measures, for both the top 5% most heavily involved bettors by total wagered, and the remaining 95% of the sample. Parametric statistics for both cohorts are not reported given their heavily skewed nature as described in the previous section.

Our results begin with substantial differences in the typical duration of play between those most heavily involved and the remaining 95% of the sample. Whilst the typical player in the majority only plays for a single day, placing approximately 9 bets in total, the typical heavily

**Table 6. Non-parametric descriptive statistics of the behavioural measures for the top 5% most heavily involved bettors by total amount wagered, and the other 95% of players.**

| Measure | Top 5% (*n* = 518) | | | Other 95% (*n* = 9, 839) | | |
|---|---|---|---|---|---|---|
| | Median | IQR | K-S | Median | IQR | K-S |
| Duration (in days) | 35 | 120 | 0.91 | 1 | 7 | 0.84 |
| Frequency | 50 | 78 | 0.98 | 100 | 50 | 0.97 |
| Number of bets | 644 | 1660 | 1 | 9 | 47 | 0.84 |
| Bets per day | 68 | 77 | 1 | 5 | 18 | 0.84 |
| ETH per bet | 1.84 | 5.61 | 0.53 | 0.10 | 0.28 | 0.50 |
| Total wagered | 986.39 | 1759.01 | 1 | 1.10 | 10.89 | 0.50 |
| Net loss | 10.3 | 102.6 | 0.56 | 0.04 | 0.6 | 0.22 |
| Percent loss | 0.9 | 7.6 | 0.38 | 6.6 | 57.6 | 0.56 |

All one sample K-S test statistic values are significant at the $p < 0.01$ level indicating the data for each measure is non-normally distributed.

involved bettor plays for over one month, placing over 600 bets. Furthermore, the typical heavily involved bettor appears to spread these bets over the month, betting approximately every other day, just under 70 times. The difference between the typical bets per day multiplied by the typical number of betting days per month (70×15 = 1050), and the typical number of bets alone (644), indicates a difference in the range of behaviours these players are exhibiting. Exploring these differences represents a key area of future work.

Each of the bets of the typical heavily involved bettor are also not insignificant in size, being almost 20 times higher than the typical player in the majority of the sample—a median 1.84 ETH (roughly $200) compared to 0.1 ETH (roughly $10). The most dramatic difference, and most concerning for the players affected, is the difference between the median total amount wagered between the most heavily involved bettors and the remaining players. With a median of almost 1,000 ETH (equivalent to approximately $100,000), it dwarfs the median 1.1 ETH $110) presented by the majority of bettors. This proportional difference is consistent with LaBrie et al's original study on regular online casino gamblers (see S1 Table), but appears to amplified in decentralised gambling application use. This difference of an almost 1000× greater total amount typically wagered by heavily involved players compared to the majority of players is a key finding of this work.

As with the behavioural measures reported for the entire sample in Table 4, the inter-quartile ranges of each of the measures leaves a wide range of potential transaction behaviours for those in the top 5%. This includes the duration measure, with players engaging with the decentralised gambling applications across a range of over 35 ± 120 days or more. This holds for the frequencies, with some heavily involved players betting every day throughout the duration of use, and some betting only a few times with large wagers. Most varied in terms of non-financially oriented measures is the number of bets placed, which presents an inter-quartile range of over 1,600 for the top 5% compared to 47 for the majority. This is of particular interest regarding the use of this data for transaction pattern analysis, a potentially fruitful area of research extending this work, and discussed in more detail below. With so comparatively few transactions made by the majority of players, further studies using this data should use behavioural measures which account for this difference.

Other widely varying measures include the total amounts wagered and the net loss. The median values of total amount wagered are 986 and 1.1 with inter-quartile ranges of 1759 and 11 respectively between cohorts. Net loss shows similar ranges with medians of 10.3 and 0.04 with inter-quartile ranges of 103 and 58 respectively. This develops the previous finding that among the top 5% of most heavily involved players, a wide range of potential patterns exist,

confirming the existing idea that there is no single behaviour indicative of heavy involvement, rather a spectrum of potential patterns and behaviours which each result in large total expenditures.

Lastly, with respect to the descriptive statistics presented, the percent loss between the most heavily involved players and the majority presents a counter-intuitive result. With a larger total amount wagered, the losses one may anticipate for the typical heavily involved individual would be high, although, in decentralised gambling applications it appears to be the opposite. With a median percent loss of just 0.9 and an inter-quartile range of 7.6, the typical heavily involved bettor does not appear to lose the amount they wager in as varied a fashion as the other 95% of the sample. These values align with Labrie et al's original work, which also reports lower percent losses for heavily involved bettors (2.5%) than for the majority of the population (5.9%). This may be an artefact of the provable fairness of these games as described in the Data Sample section above, where players can be certain of the amount the 'house' is taking from each bet, or it may be a result of extensive repeated play, where the range of potential losses is effectively smoothed by the larger sample available for each player. In the case of the majority of players, a median percent loss of 6.6 and inter-quartile range of 57.6, suggests large relative wins and losses for the relatively small bets they place. This finding differs from the original work, but makes logical sense given the non-commital and non-intense behaviours described above for the typical player of decentralised gambling applications.

The one sample K-S statistics reported for the behavioural measures of the heavily involved portion of players and the remaining 95% indicate several measures of interest for future work. Specifically, the differences between the first four measures (duration, frequency, bet count, and bets per day) do not appear substantially different from one another. The differences in distributions between the total amounts wagered however are vastly different, with a coefficient of 1.00 (to 2 decimal places) for heavily involved bettors compared to 0.50 for the majority. This may be a fruitful area of further exploration, as the underlying distributional differences for these measures may be used in conjunction with other measures to predict heavy involvement.

## Discussion

This study presents the first ever analysis of decentralised gambling transactions on the Ethereum blockchain. Decentralised gambling, and the contract components of their architectures, present significant regulatory challenges [1], whilst simultaneously offering rich transaction level data for research. Whist this transaction level data exists in large quantities, we have shown that the entire set is not immediately useful for research given the likely presence of non-human players. This means that although a large, publicly available, *in-vivo* data source for player behaviour research has emerged, scholars must take care when using it to solve existing problems, especially when exploring issues around disordered transaction patterns and player behaviour clustering.

### Non-human players in decentralised gambling applications

Our first distinct analysis involved employing statistical tests to detect differences between transactions of (likely) human and non-human origin. To this end, we found that performing two sample Kolmogorov-Smirnov tests between behavioural measures, and between games provided by decentralised gambling applications, can be effective for detecting the presence of players whose transactions stand out against those in other games. This simple method invites improvements, but shows that relying on distributional differences between human and non-human players is enough for meaningful distinction at this early stage.

Importantly, we hold the assumption that of the nine application-game combinations we explored, the one that stands out as different is not being transacted with by human players, as opposed to the other way around. Under this assumption, we may suggest that the reason it differs so substantially from others is that the majority of the players are in fact not human, but instead are cryptocurrency spending/betting algorithms designed to transact with the application, potentially to inflate perceived popularity. Exploring motivations behind algorithmic trading with these applications presents an interesting but tangentially related area of future work.

## Cryptocurrency gamblers and behavioural relationships

The second and third analyses described in the Present Study section above aimed to describe the gambling behaviours of users of decentralised gambling applications, and assess the relationships between these measures. Our results suggest, as with similar existing work, that the distributions of all behavioural measures are significantly skewed, and therefore benefit from the application of non-parametric statistics. Applying such statistics, and without break-ing down the sample of players into meaningful sub-samples, the typical user of decentralised gambling applications does not appear to be heavily involved, and does not appear to place a substantial number of high of bets. However, this description fails to capture the most important aspect of the findings in this study, which are that those most heavily involved in the use of decentralised gambling applications appear to spend significantly more than both the majority of the population, and more than heavily involved gamblers in other types of online gambling. Exploring this relationship further and breaking down differences in terms of the behavioural measures calculated for each player, presents a fruitful area of further work if findings building on previous studies are to be translated to this new domain.

Furthermore, this study's design draws heavy inspiration from early work describing online casino game players. The data available to the original researchers took a daily aggregate form. This means that the behavioural measures they devised to describe cohorts of players perhaps do not capture the depth of insight available when using individual transaction level data as available via cryptocurrency transactions. There may therefore be behavioural measures which appear inaccessible at the daily aggregate level, such as average gambling session length or average rate of spending. To the author's knowledge, studies in the field of player behaviour tracking have not yet explored such granular measures, nor applied them to data sets across different types of online gambling. That considered, measure-oriented work such as Kainulai-nen's [15], which describes a new measure of risk taking specific to gambling, presents the opportunity to apply new techniques to gain deeper insight on player behaviours.

## Heavily involved cohort characteristics

Our final analysis aimed to provide an epidemiological description of the gambling behaviours of an empirically determined group of heavily involved gamblers. The results regarding this cohort of players, identified as heavily involved by total amount wagered, suggest a number of important discoveries. Firstly, although the typical heavily involved player spends the equivalent of over $120,000 during a 35 day period, the losses they typically incur as a percentage of their amount wagered are under 1%. This means that although their expenses dwarf the majority of players by over 1000×, they do not appear to be losing as much proportionally as the majority of players, who, when placing approximately $105 worth of bets in total over a one day period typically lose under 6%, or $7. It is important to note that this difference in losses between heavily and non-heavily involved players is not unique to decentralised gambling applications, as evidenced by LaBrie et al's original findings (S1 Table).

Another important result of the analysis regarding heavily involved bettors is the typical difference in bet size, with heavily involved players wagering just under 20× more than their low to moderate counterparts. This result can be used to inform further research on the use of cryptocurrencies for gambling, and the analysis of their transactions, for the early detection of unsustainable spending, for example. This is just one of many possible—and much needed—avenues of work extending these findings into the domain of responsible gambling analytics.

## Limitations

The analyses performed here are subject to many of the same limitations of the use of online gambling data for behaviour tracking research generally [5]. These include issues surrounding the generalisability of findings. In the context of the use of cryptocurrencies for gambling—specifically through decentralised gambling applications—is unclear whether the analysis undertaken here will have similar results across other comparable applications. Furthermore, it may be the case that the behavioural patterns uncovered here are incomplete as true player gambling behaviour may be spread across several unobserved applications in addition to the applications discussed here.

Such fundamental limitations can not be completely negated through experimental design. However, future work should focus on increasing the sample size, both to more applications and more players, which may address the issue of generalisablity regarding decentralised gambling applications.

An additional point may be made regarding the transaction matching process performed which pairs incoming and outgoing transactions. The data that this analysis was conducted over involves a complete record of each player's ingoing and outgoing transactions. However, it does not contain a reliable temporal ordering for this data. In order to create a more useful data set than the incoming and outgoing transactions in isolation, they can be matched such that an outgoing transaction chronologically following an incoming transaction from the same address can be taken to be the payout of a previously placed bet, but that other candidate transactions may be considered in the case that one transaction is completed ahead of another.

Transaction matching, as described above, is unnecessary for the methodology used in this paper, as all behavioural measures computed use the aggregation of an individual's ingoing and outgoing transactions. For example, the behavioural measure of percentage loss for a given player only requires the sum of their bets and the sum of their payouts. However, one might imagine the calculation of more sophisticated behavioural measures that do require matched data in order that more sophisticated analyses might take place. For example, one might attempt to calculate the phenomenon of 'chasing losses' by measuring the extent to which players place larger bets after losing money on a prior bet. Such an analysis is not possible using the data set outlined above, as any given payout could not be conclusively matched to a single bet. This matching process is briefly mentioned here as it will be essential for future work in this area at the individual bet and risk analysis levels—both techniques are considered out of scope of the present study.

Other limitations relate to the nature of the applications themselves in comparison to other online gambling platforms. Specifically, each of the applications used here—and all decentralised applications atop cryptocurrency networks—must use cryptocurrencies or similar tokens by design. This means that although the real world value (e.g. in US$ or GBP) for any amount of cryptocurrency can be determined in real terms, it is unclear whether or not this relationship affects wagering, and in what way. An area of future work exploring this relationship may investigate the distributions of bet sizes, and may uncover more detailed findings in terms of how decentralised gambling differs from other online gambling. These studies may also help

in understand how the use of virtual goods and currencies affects the behaviour of players with respect to spending. In this vein, comparisons with other uses of cryptocurrency technology, such as the development of crypto-games [16], may provide a useful basis for comparison.

The differences between these applications and other online gambling providers also inherently affects the populations who use them. This means that the sample of players considered in this work is a sub-population of individuals who have purchased cryptocurrencies—a volatile [16] and technologically sophisticated means of facilitating e-commerce [1]. How the personal characteristics of this sub-population differ from the general population is largely unknown—especially with respect to gamblers—and represents an important avenue of future work.

A final limitation of this work, given the context of recent advances in player behaviour tracking research, is that it only explores simple behavioural measures based on those used to explore casino gamblers. It therefore does not reach into more advanced analytical methods for describing, classifying, or predicting player behaviours. This includes work by Fiedler which explores more granular behavioural measures [6, 11], multiple studies by Percy [13] and Dragičević *et al.* [17], which employ neural networks and other machine learning methods for responsible gambling, and other data mining procedures for identifying high risk gamblers as done by Philander [14]. In order to apply supervised machine learning as in these studies, labelling heuristics for players should also be explored.

## Future work

The analysis presented here recreates that of a series of papers originating from Harvard Medical School [9, 10]. Since that series was first published, a number of other descriptive measures have been used such as the intensity, variability, frequency, and trajectory of a player's bets [3], and more specific variables such as the number of betting sessions and total time spent betting [11]. Extending the present study by exploring player behaviours across these dimensions would give a more complete picture of the player base of decentralised gambling applications, and would give stronger grounds on which these transactions may be compared with other types of gambling.

A second avenue of research extending the descriptive and test statistics reported here is the use of this data for identifying and predicting high risk gambling. Existing work has identified transaction patterns and behaviours to be markers of high-risk play [4, 18]—exploring such methods in this new domain may therefore help identify those at risk, and better describe the way these applications are used. The development of such identification methods may spur regulators and policy makers to further explore cryptocurrency exchanges, whose operations provide financial access to these applications. An obvious and useful first step would be formally requiring transaction reporting for responsible gambling analysis.

Finally, the findings presented here may be tentatively mapped to other forms of gambling in which similar work has been reported. Generalisations drawn from such mappings may require further data gathering from both additional cryptocurrencies, such as the EOS network, and more applications on the Ethereum network as described in this paper and elsewhere. Increasing the sample size of players, both human and otherwise, represents a strong second step in creating reliable and generalisable findings, which extend this work.

## Conclusion

In this study, 2, 232, 741 transactions to and from three decentralised gambling applications, originating from 24, 234 unique cryptocurrency addresses, were gathered, and four distinct analyses performed. Our findings suggest that not all transactions to decentralised gambling

applications originate from human players, making data cleaning crucial in all further academic work concerning this type of data. We found a pairwise two sample Kolmogorov-Smirnov test across players behavioural measures to be effective in distinguishing non-human players. Of transactions believed to originate from human players, we found that the behavioural measures computed naïvely describe non-intensive but moderate spending over a short duration for the typical player. This description was then found to mask a small portion of heavily involved bettors, whose typical bet size appears to be almost 20× larger than their non-heavily involved counterparts, and their total amount wagered appears to be over 1000× larger over their duration of play. Our contributions in this paper are two-fold; the work presented primarily illustrates the power and scale of transaction data that decentralised gambling applications can provide gambling researchers. Secondly, it describes a large cohort of players from three such applications, and uncovers extreme behaviours, such as large bet sizes and substantially larger total wagering among heavily involved players. This work should draw attention to cryptocurrency transactions as a tool for large scale *in-vivo* gambling research, and presents a robust foundation upon which multiple avenues of further analyses can be performed.

## Supporting information

**S1 Table. Casino game player description.** The behaviours of the top 5% and other 95% of casino bettors by total amount wagered, reprinted with no modifications from LaBrie et al's 2008 study on casino game players [1], on which the present study's methodology is based. It is included here following reviewers recommendations for comparison.
(PDF)

## Author Contributions

**Conceptualization:** Oliver J. Scholten.

**Data curation:** Oliver J. Scholten.

**Methodology:** Oliver J. Scholten, David Zendle, James A. Walker.

**Project administration:** Oliver J. Scholten, James A. Walker.

**Resources:** Oliver J. Scholten.

**Supervision:** David Zendle, James A. Walker.

**Validation:** Oliver J. Scholten.

**Visualization:** Oliver J. Scholten.

**Writing – original draft:** Oliver J. Scholten.

**Writing – review & editing:** Oliver J. Scholten, David Zendle, James A. Walker.

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
