## [Decision Letter · Decision Letter 0]

30 Jul 2020

PONE-D-20-20436

Inside the Decentralised Casino: A Longitudinal Study of Actual Cryptocurrency Gambling Transactions

PLOS ONE

Dear Dr. Scholten,

Thank you for submitting your manuscript to PLOS ONE. After careful consideration, we feel that it has merit but does not fully meet PLOS ONE’s publication criteria as it currently stands. Therefore, we invite you to submit a revised version of the manuscript that addresses the points raised during the review process.

We look forward to receiving your revised manuscript.

Kind regards,

He Debiao

Academic Editor

PLOS ONE

Journal Requirements:

2. Thank you for providing URLs to the various websites and data repositories used in the study. However, PLOS ONE does not allow for footnotes in its publications; as such, we ask that you move all of the footnotes to the main text.

Reviewers' comments:

Reviewer's Responses to Questions

**Comments to the Author**

1. Is the manuscript technically sound, and do the data support the conclusions?

Reviewer #1: Yes

Reviewer #2: Yes

Reviewer #3: Yes

2. Has the statistical analysis been performed appropriately and rigorously? 

Reviewer #1: Yes

Reviewer #2: Yes

Reviewer #3: Yes

3. Have the authors made all data underlying the findings in their manuscript fully available?

Reviewer #1: No

Reviewer #2: Yes

Reviewer #3: Yes

4. Is the manuscript presented in an intelligible fashion and written in standard English?

Reviewer #1: Yes

Reviewer #2: Yes

Reviewer #3: Yes

5. Review Comments to the Author

Reviewer #1:

Minor:

1.     Improve language especially in abstract.

2.     Include graphical diagrams to enhance understandability.

3.     A comparative graph or table with existing analysis of similar type on similar apps have to be included in result portion.

Major: The statistics of data collected from application like Etheroll, and Dice2Win is not shown and the span over which data is collected.

1.     The summary of data and the periods over which data has been collected require a graphical representation before the final analysis is presented.

2.     The design of algorithm/method which conducted the analysis is not shown in any form (textual or graphical).

3.     Analysis required to be interlinked with the suggested improvements and detected issues which can help improve the online gambling through games. The Author has to lead readers to an application analysis.

Reviewer #2:

This study explores decentralized gambling that users use Ethereum cryptocurrency to cover their identities. They analyzed 2,232,741 transactions from 24,234 unique addresses. The results of study shows that the use of these applications as a research platform, specically for large scale longitudinal in-vivodata analysis.

Reviewer #3:

The research article tracks the users' behavior in decentralized gambling applications. The domain of application is considered new and promising. The data was sufficient to apply behavioral measures to reach the conclusion and the statistical analysis has been performed appropriately. The manuscript was well written.

---

## [Author Response · Author response to Decision Letter 0]

14 Sep 2020

We would like to thank the reviewers for their time, valuable suggestions, and constructive comments! We also believe the scholarly potential of this new area to be very exciting, and have made several improvements to address your comments. We specifically worked to improve the clarity of the paper and the analysis it presents.

Please find each of the issues raised below, followed by a summary of our improvements;

1. Improve language especially in abstract [R1]

We have improved the language throughout by removing complex phrases, simplifying sentence structures in all sections, and improving the overall readability of the paper throughout. All of our changes are highlighted in the tracked changes document for your convenience. Special attention has been given to the abstract which is now more accessible.

2. Include graphical diagrams to enhance understandability [R1]

Two new figures have been added to enhance understandability, the first visualises the periods over which the data have been collected, and the second shows summary statistics for the data collected in graphical form. These diagrams together bring much needed clarity to the scope of the data collection performed, and provide the reader with a more intuitive understanding of the scale of our investigation.

3. A comparative graph or table with existing analysis of similar type on similar apps have to be included in results portion [R1]

Given the youth of the technology used in these apps, no such analysis of similar type exists in the current literature so this is currently not possible. This considered, we have included a table from the study on which our methodology is based, which also targets online (though not decentralised) casino game players for comparison, and have added some comparative statements in the results portion to better frame our findings. 

4. The statistics of data collected from applications like Etheroll, and Dice2Win is not shown and the span over which data is collected; The summary of data and the periods over which data has been collected require a graphical representation before the final analysis is presented. [R1]

Both diagrams added to address (2) add to better describing the data collected as part of this research. On top of this, a table has been added and referenced in the Data Sample section presenting a number of statistics to more comprehensively describe the data collected from these applications. This table combined with the new figures comprehensively describe the dataset, which will be available through an OSF repository linked throughout.

5. The design of algorithm/method which conducted the analysis is not shown in any form (textual or graphical) [R1]

To make our analysis clearer we have included pseudo-code of the bot-detection procedure, and have added sentences throughout highlighting our fully documented and open source code which is available on github. Our study is fully and quickly replicable using the library referenced, and we invite our readers to do so by providing full access to our data and code.

6. Analysis required to be interlinked with the suggested improvements and detected issues which can help improve the online gambling through games. The author has to lead researchers to an application analysis. [R1]

We have integrated the changes outlined above into a more consistent narrative, but must stress that this paper’s aim is not to improve online gambling through games, but rather to improve the study of player behaviours using the new paradigm of cryptocurrency transaction data. We have clarified this in the introduction, emphasising that the application analysis this paper presents is the first step towards developing a better understanding of players, and therefore lays the foundation for more advanced analytical methods in this advanced and emerging domain. The youth and lack of existing work in this area is also stressed in the introduction, better leading the reader to the reason for our application analysis.

We hope our revisions fully address your feedback, and would like to thank all reviewers for their encouraging comments to improve our paper.

---

## [Decision Letter · Decision Letter 1]

1 Oct 2020

Inside the Decentralised Casino: A Longitudinal Study of Actual Cryptocurrency Gambling Transactions

PONE-D-20-20436R1

Dear Dr. Scholten,

We’re pleased to inform you that your manuscript has been judged scientifically suitable for publication and will be formally accepted for publication once it meets all outstanding technical requirements.

Kind regards,

He Debiao

Academic Editor

PLOS ONE

Additional Editor Comments (optional):

Reviewers' comments:

Reviewer's Responses to Questions

**Comments to the Author**

1. If the authors have adequately addressed your comments raised in a previous round of review and you feel that this manuscript is now acceptable for publication, you may indicate that here to bypass the “Comments to the Author” section, enter your conflict of interest statement in the “Confidential to Editor” section, and submit your "Accept" recommendation.

Reviewer #1: All comments have been addressed

Reviewer #3: All comments have been addressed

2. Is the manuscript technically sound, and do the data support the conclusions?

Reviewer #1: Yes

Reviewer #3: Yes

3. Has the statistical analysis been performed appropriately and rigorously? 

Reviewer #1: Yes

Reviewer #3: Yes

4. Have the authors made all data underlying the findings in their manuscript fully available?

Reviewer #1: Yes

Reviewer #3: No

5. Is the manuscript presented in an intelligible fashion and written in standard English?

Reviewer #1: Yes

Reviewer #3: Yes

6. Review Comments to the Author

Reviewer #1: Authors have already addressed all the concerns. Overall manuscript structure has been enhanced and readibility has been improved significally

Reviewer #3: Thank you for considering my comments. The manuscript became better after the modifications.

However, the data underlying the findings has not been made available. As it sends an access request to the owner.

7. PLOS authors have the option to publish the peer review history of their article (what does this mean?). If published, this will include your full peer review and any attached files.

Reviewer #1: **Yes: **Malik Muhammad Ali Shahid

Reviewer #3: **Yes: **Randa Aljably

---

## [Editor Report · Acceptance letter]

6 Oct 2020

PONE-D-20-20436R1 

Inside the decentralised casino: a longitudinal study of actual cryptocurrency gambling transactions 

Dear Dr. Scholten:

I'm pleased to inform you that your manuscript has been deemed suitable for publication in PLOS ONE. Congratulations! Your manuscript is now with our production department. 

Kind regards, 

on behalf of

Dr. He Debiao 

Academic Editor

PLOS ONE